# Molecular Modeling Studies to Probe the Binding Hypothesis of Novel Lead Compounds against Multidrug Resistance Protein ABCB1

**DOI:** 10.3390/biom14010114

**Published:** 2024-01-16

**Authors:** Yasmeen Cheema, Kenneth J. Linton, Ishrat Jabeen

**Affiliations:** 1School of Interdisciplinary Engineering & Sciences (SINES), National University of Science and Technology, Sector H-12, Islamabad 44000, Pakistan; ycheema.phdcse16@rcms.nust.edu.pk; 2Blizard Institute, Faculty of Medicine and Dentistry, Queen Mary University of London, London E1 2AT, UK; k.j.linton@qmul.ac.uk

**Keywords:** ABCB1, molecular docking simulation, MD simulation, ligand–ABCB1 interactions

## Abstract

The expression of drug efflux pump ABCB1/P-glycoprotein (P-gp), a transmembrane protein belonging to the ATP-binding cassette superfamily, is a leading cause of multidrug resistance (MDR). We previously curated a dataset of structurally diverse and selective inhibitors of ABCB1 to develop a pharmacophore model that was used to identify four novel compounds, which we showed to be potent and efficacious inhibitors of ABCB1. Here, we dock the inhibitors into a model structure of the human transporter and use molecular dynamics (MD) simulations to report the conformational dynamics of human ABCB1 induced by the binding of the inhibitors. The binding hypotheses are compared to the wider curated dataset and those previously reported in the literature. Protein–ligand interactions and MD simulations are in good agreement and, combined with LipE profiling, statistical and pharmacokinetic analyses, are indicative of potent and selective inhibition of ABCB1.

## 1. Introduction

Intrinsic or acquired resistance to chemotherapy, known as multidrug resistance (MDR), is an obstacle in the treatment of cancer [1,2,3]. Several MDR-causing factors have been elucidated; however, accelerated drug efflux mediated by the overexpression of the ATP-binding cassette (ABC) transporters is commonly recognized as clinically important [4,5,6]. Among the 49 distinct human ABC transporters, ABCB1 is the most commonly identified mediator of MDR and can therefore be considered a key therapeutic target [7,8,9]. Over the past few decades, three generations of ABCB1 inhibitors have been developed to overcome the innate or upregulated ABCB1 expression. Yet, to date, no single drug has been approved due to a combination of poor pharmacokinetic properties, a lack of selectivity, interpatient variability and poor toxicity profiles [10,11,12,13]. However, we are now in a new era of understanding of the transporter structure and function of several structures of the human ABCB1, which were determined via cryo-electron microscopy. The first was solved in the ATP-bound conformation (pdb: 6C0V [14]), and several have been solved with transport substrates (pdb: 6QEX [15] and 7A69 [16]) or inhibitors bound (pdb: 7A6E, 7A6C [16], and 7O9W [17]). An induced-fit mechanism is thought to explain the poly-specificity of the transporter, which may complicate the use of models for inhibitor design. The other complicating factor is that the structures solved with transport substrates or inhibitors are also trapped by inhibitory antibodies, which could mean that the binding pockets may not be optimal. Nevertheless, their utility for the identification of more potent, specific and less toxic ABCB1 inhibitors to combat MDR is being explored and tested by ourselves and others [18,19]. Previous structural optimization of ABCB1 inhibitors suggests that high molecular weight, lipophilic efficiency (LipE), large logP values, longer skeletal structure and the accumulation of more hydrophobic groups increase the binding affinity of inhibitors [20]. We have also applied structure/activity analyses to identify novel lead compounds denoted as ‘A’, ‘D’, ‘E’ and ‘F’ (Figure 1) [21]. Their potency and efficacy were proven by their ability to re-sensitize ABCB1-expressing cells to the anticancer drug taxol. The compounds were tested against the Lipinski, Ghose, Veber, Egen and Mugge filters.

In the present study, a curated dataset of structurally diverse and selective inhibitors of ABCB1 was used to probe the structure activity relationship to guide molecular modelling outcomes. The dataset of 58 inhibitors was docked into the binding pocket of human ABCB1 (6QEX), followed by ligand–protein interaction analysis, structure/activity relationships (SAR)-guided pose analysis, protein–ligand interaction fingerprints (PLIF), ligand–protein contacts analysis, statistical analysis and a molecular dynamics simulation study for 100 ns. The data highlight the possibility to develop more potent and specific inhibitors of the transporter while minimizing off-target effects.

## 2. Materials and Methods

### 2.1. Dataset Collection

Four recently identified novel lead compounds [21] with biological activity ranging from 0.001 to 0.019 µM were used to probe the binding hypothesis within ABCB1. An additional dataset of 54 structurally diverse compounds that are purportedly selective compounds for inhibition of P-gp/ABCB1 (over ABCG2 and ABCC1) was obtained from the literature [22,23,24] for the comparison of the binding hypothesis. Of the 54 compounds, 16 were highly selective and active towards ABCB1 in comparison with both ABCG2 and ABCC1, as shown in Appendix A, whereas 19 compounds were more active against ABCB1 as compared to ABCC1 (Appendix A) and a further 19 were active against ABCB1 as compared to ABCG2 (Appendix A). The dataset belongs to four structurally diverse classes: quinolone, pyrogallol, quinazolinone and propafenone [22,23,24,25], and the reported experimental inhibitory potency (IC_50_) values of the dataset against ABCB1 were in the range of 0.001 to 113 µM.

### 2.2. Molecular Docking Simulation, SAR and Pose Analysis

To analyze the ligand–protein interaction pattern and to obtain deeper insights into the binding pattern, a molecular docking simulation of the complete dataset of ABCB1 inhibitors including the four lead compounds was performed using GOLD [26] software (5.6.1). Briefly, the electron-microscopic, nanodisc-reconstituted structure of human ABCB1 structures solved to 3.6 Å bound to the inhibitory antibody UIC2 fab and with taxol in the binding pocket (6QEX) [15] was retrieved from the Protein Data Bank (PDB) [27] for the molecular docking simulations. The general energy minimization protocol was applied to minimize the structure of ABCB1 using default parameters and the Amber10 force field [28].

The structure of the whole dataset was generated using ChemDraw Ultra 12.0 [29] and then cleaned using Accelrys Draw 4.1 software [30]. The inhibitors were protonated by titrating all the atoms using the Merck Molecular Force Field 94x with energy minimization in 80% solvent at pH 7.4 [31]. The complete transmembrane region of ABCB1 was considered the binding pocket [17] of the protein and 100 poses of each compound within the binding pocket were generated using the Gold score function [26]. The docked protein–ligand complexes were analyzed using Protein–Ligand Interaction Profiler (PLIP) [32] and Pymol software [33] (2.5.1) followed by investigation of the interactions between the protein and the binding solutions of the whole dataset using the protein–ligand interaction fingerprint (PLIF) in MOE. To perform the PLIF analysis, 100 conformations of each compound of the dataset were given as input to identify the interaction frequency of the ligands and the receptor protein followed by the generation of a histogram of the most frequent interactions. Pose analysis was guided by SAR for the three structurally diverse classes. Consequently, a thorough comparison of interactions of the four docked leads with the docked dataset of 54 compounds was performed on the basis of their PLIF- and SAR-guided pose analyses.

### 2.3. Calculation of Physicochemical Parameters and Statistical Analyses

To guide the analysis of the molecular modeling study, 2D descriptors such as the molecular weight, number of heavy atoms and clogP of all the compounds were calculated using the descriptor calculation in Molecular Operating Environment (MOE) [34] and Bio-Loom software [35] (1.5). The lipophilic efficiency (LipE) values of the dataset were also calculated according to Equation (1), where clogP and pIC_50_ represent the lipophilicity and biological potency of the compounds, respectively.
LipE = pIC_50_ − clogP(1)

The whole dataset of compounds along with their pIC_50_, clogP, molecular weight (MW), IC_50_ and LipE values are presented in Appendix A. Principal component analysis (PCA) was executed using R 3.5.0 (R Core Team, 2018) to analyze the collinearity between the Gold score, pIC_50_ and clogP values of the whole dataset [36]. To obtain the eigenvectors (PCs) and the eigenvalues (the weight of each PC), the covariance matrix was generated followed by the diagonalization process.

### 2.4. Molecular Dynamic Simulation

The stability analysis of the statistically significant, top-scoring ligand–protein complexes of the lead compounds (‘A’, ‘D’, ‘E’ and ‘F’) with suitable binding interactions and LipE and clogP values was performed using the Desmond package (Schrodinger) [37]. To achieve the thermodynamic stability of all the ligand–protein complexes, the systems were prepared using rectification and refinement of proteins, system building and energy minimization followed by an MD run of 100 ns.

The Schrodinger protein preparation wizard was used to add hydrogen bonds, for remodeling the tautomeric and ionization states of amino acid residues and creating disulfide bonds, followed by energy minimization using the OPLS4 force field. The system was built in Schrodinger with the Transferable Intermolecular Interaction Potential 3 Points (TIP3P) solvent model with an orthorhombic box shape and OPLS4 force field. The membrane model of phosphatidylcholine (POPC) was used to place the membrane automatically. POPC is a common phospholipid in biological membranes and is added to explore the lipid bilayer constraints on protein movement and membrane–protein interactions during MD simulation. To ensure the selection of transmembrane atoms, the amino acids within the binding pocket were provided explicitly and their intersection with helices, loops and strands of the secondary structure of ABCB1 was also ensured. To replicate the physiological circumstances first, ions were excluded from the simulation box but then salt molecules were positioned to initialize the simulation. Ion placement was then recalculated by neutralizing sodium ions (Na^+^) and adding 0.2 M of potassium and chlorine ions (K+ and Cl^−^) in the OPLS4 force field. Finally, the MD simulation was performed using the isothermal–isobaric (NPT) ensemble for all selected ligand–protein complexes; atmospheric pressure was maintained at 1.0 bar and the temperature was set to 303 K. To study the dynamics and stability of docked complexes, trajectories were recorded with approximately 250 frames for 100 ns. The average change in displacement of atoms with respect to the reference frame was measured and, to gain insights into the structural conformation throughout the simulations, the root mean structure deviation (RMSD) was calculated using Equation (2).
(2)$$\mathrm{RMSD}=\sqrt{(1/N)\sum_{i=1}^{N}(r\left(t_{x}\right))-(r(t_{ref}{))}^{2}}$$
where: *N* is the number of atoms in the atom selection.*t_ref_* is the reference time (typically the first frame is used as the reference and it is regarded as time *t* = 0).*r* is the position of the selected atoms in frame *x* after superimposing on the reference frame, where frame *x* is recorded at time *t_x_*.

The root mean square fluctuation (RMSF) was calculated to characterize the local changes along the protein chain using Equation (3).
(3)$$\mathrm{RMSFi}=\sqrt{(\frac{1}{T})\sum_{t=1}^{N}<(r_{i}\left(t\right))-(r(t_{ref}{))}^{2}}>$$
where:*T* is the trajectory time for the RMSF.*t_ref_* is the reference time.*r_i_* is the position of residue *i*.*r* is the position of atoms in residue *i* after superposition on the reference.

The angled brackets represent the average of the square distance taken over the selection of atoms in the residue.

### 2.5. Pharmacokinetic Analysis

To gain insights into the potential toxicity and drug-like properties of the lead com- pounds, pharmacokinetic analysis was performed using webservers SwissADME (2.6.0) [38], Pro-Tox II (2.0) [39] and Molinspiration (2023.01) [40]. These tools are used to predict the toxicity, absorption, distribution, metabolism and excretion (ADME), as well as the physicochemical properties of the compounds. The full workflow for the integrated analyses is described in Figure 2.

## 3. Results

### 3.1. Molecular Docking Simulation

The dataset of structurally diverse ABCB1 inhibitors was docked into the transmembrane domain of the cryoEM structure of human ABCB1 (PDB: 6QEX) using GOLD 5.3 suit software [26]. The binding region was delineated by those amino acids that were residing within a radius of 30 Å from X = 171, Y = 166 and Z = 169, which comprises the whole transmembrane domain region. A comparative analysis of the protein–ligand interaction of the four lead compounds [21] with the docking results of 54 selective compounds [22,23,25,41] was followed by PLIF and SAR-directed pose analysis. All four lead compounds ‘A’, ‘D’, ‘E’ and ‘F’ docked in the transmembrane region and mostly formed hydrophobic interactions with Phe-303, Tyr-307, Phe-343, Met-986 and Gln-990, π–π interactions with Trp-232 and hydrogen bonding with Gln-990, as visualized in Figure 3.

### 3.2. Molecular Dynamics Simulations of Leads

To obtain deeper insight into the conformational flexibility and stability of lead compounds ‘A’, ‘D’, ‘E’ and ‘F’ within human ABCB1, molecular dynamics (MD) simulations of 100 ns were performed. The very low positional deviation of the RMSD of the lead compounds complexed with ABCB1 (maroon) indicates the binding stability of the compounds within the protein. Figure 4, panel 1 shows the RMSD of the apo-ABCB1 in blue and the ABCB1-compound ‘A’ complex (ABCB1-A) in maroon. The RMSD of the apo-ABCB1 varies from 2.5 Å to 3.8 Å with an average value of 3.1 Å. In comparison, the RMSD of the ABCB1-A complex fluctuates between a narrower range of 3.5 Å to 3.8 Å with an average value of 3.6 Å throughout the simulation period of 100 ns, which reflects the structural stability of ABCB1 after binding with lead compound A. Previous studies have also reported that fluctuations of the order of 1–3 Å are perfectly acceptable for bimolecular systems [42]; however, in the case of ABCB1, slightly higher fluctuation can also be considered due to its larger size [43].

Similarly, the 100 ns simulation trajectory for ABCB1 in complexes with compounds ‘D’, ‘E’ and ‘F’ in Figure 4, panels 2, 3 and 4 respectively, showed negligible fluctuation in the RMSD of the protein as well as the docked lead compound ‘D’ after the first 15 ns. 

The effect of the binding of the compounds on the internal dynamics of the ABCB1 was examined in the fluctuation of each atom around its average position via the RMSF during the 100 ns simulation. Overall, the RMSF plot of all the complexes (Figure 5) fluctuated by less than 3 Å; relatively large fluctuations were observed in the loop regions as expected, due to their flexible nature. Comparatively small peaks were observed in the transmembrane regions as well as the Walker A and B regions (schematic presentation is provided is Appendix A). The RMSF of the protein backbone showed similar behavior for all of the simulated complexes and the low movement of both nucleotide-binding domains (NBD1 and NBD2) around 3 Å that might suggest inhibition of the transport cycle [43,44], which can further be validated by ATPase assay. Thus, all four of the docked lead compounds ‘A’, ‘D’, ‘E’ and ‘F’ remained occluded in a stable binding pocket. The secondary structure elements of the protein (SSE) in the presence of all four lead compounds were also maintained over the simulation period, further suggesting the structural integrity of the complexes. Complexes ‘ABCB1-A’, ‘ABCB1-D’, ‘ABCB1-E’ and ‘ABCB1-F’ maintained 49.05%, 56.94%, 55.00% and 55.23% alpha helices, respectively, and 8.59%, 9.22%, 8.91% and 9.36% beta-strands, respectively, throughout the simulation period of 100 ns, as shown in Appendix A.

To compare the stability of the novel lead compounds ‘A’, ‘D’, ‘E’ and ‘F’ with the two most active and selective compounds of the dataset, MD simulation of complexes ‘35E-ABCB1′ and ‘44I-ABCB1′ followed by the RMSD, RMSF and SSE analysis was performed. MD simulations of 35E showed the shifting of the ligand RMSD at ~55 ns from 1.2 A° to 2.4 A° due to its conformational variability within the binding pocket of ABCB1. After approximately 55 ns, the complex of 35E-ABCB1 attained stability till the end of the simulation period of 100 ns (Appendix A, panel A). In the case of compound ‘44I’, the RMSD of the apo-ABCB1 varied from 2.0 Å to 3.9 Å with an average value of 2.9 Å. In comparison, the RMSD of the ABCB1-44I complex fluctuated within the range of 2.1 Å to 3.5 Å with an average value of 2.8 Å throughout the simulation period of 100 ns (Appendix A, panel D). Moreover, the RMSF plots for both ‘35E’ and ‘44I’ show very little fluctuation in the NBD region, as shown in Appendix A, panels B and E. Furthermore, complexes ‘ABCB1-35E’ and ‘ABCB1-44I’ maintained 55.81% and 57.1% alpha helices, respectively, and 8.59% and 8.94% beta-strands, respectively, throughout the simulation period of 100 ns as shown in Appendix A, panel, C and F. Comparison of MD results of leads (‘A’, ‘D’, ‘E’ and ‘F’), ‘35E’, and ‘44I’ illustrated comparatively less fluctuation in RMSD and RMSF plots of ‘A’, ‘D’, ‘E’ and ‘F’. Moreover, after docking with lead compounds, ABCB1 maintained same structural integrity of ABCB1 consistent with the most active and selective compounds ‘35E’ and ‘44I’ as shown in Appendix A, panel A and D.

### 3.3. Protein–Ligand Interaction Analyses of Lead Compounds Pre- and Post-MD Simulations

To elucidate the molecular basis of protein–ligand interactions, the docked complexes were analyzed at 0 ns and 100 ns of the MD simulations using Protein–Ligand Interaction Profiler (PLIP) [32] and Pymol software (2.5.1) [33] (Figure 6). Compound ‘A’ was coordinated by six interactions on docking: hydrophobic interactions with Leu-65, Ile-340, Phe-343 and Ala-871 and hydrogen bonding interactions with the N-H group of the peptide backbone of Ala-871 and the side chain of Glu-875 at 0 ns, as shown in Figure 6, panel 1. All six interactions remained intact after 100 ns as presented in Figure 6, panel 5, and summarized in Appendix A.

Compound ‘D’ showed hydrophobic interactions with Phe-336, Leu-339, Ile-340 and Phe-983, and hydrogen bonding with Gln-725 and Gln-990, while it was also engaged in *π* stacking with Phe-303 as shown in Figure 6, panel 2. After 100 ns, the same interactions were observed as shown in Appendix A and Figure 6, panel 6. Similarly, for ‘ABCB1-E’ and ‘ABCB1-F’, all interactions remained intact as displayed for compound ‘E’ in Figure 6, panel 3 at 0 ns and panel 7 at 100 ns, and for compound ‘F’ in Figure 6, panel 4 for 0 ns and Figure 6, panel 8 for 100 ns and listed in Appendix A.

To answer whether the interactions between the lead compounds and human ABCB1 remained stable in the post-docking MD simulation, we analyzed the interacting residues of leads ‘A’, ‘D’, ‘E’ and ‘F’ in protein–ligand contacts in each trajectory frame during the MD simulation of 100 ns, as shown in Appendix A. Briefly, most of the interactions remained stable as they are represented by a darker shade of orange (Appendix A). In particular for lead compound ‘A’, Leu-65, Ile-340, Phe-343 and Glu-875 showed interactions in most of the trajectory frame (Appendix A, panel 1) during the simulation period of 100 ns. Lead compound ‘D’ displayed more stable interactions with Phe-303, Ile-340, Gln-725, Phe-728, Phe-983 and Gln-990 (Appendix A, panel 2) during 100 ns. Likewise, lead compound ‘E’ showed more stable interactions with Phe-303, Tyr-307, Phe-343 and Glu-875 (Appendix A, panel 3) during the simulation period. Moreover, lead compound ‘F’ displayed more stable interactions with Phe-303, Ile-306, Phe-343, Gln-347, Glu-875 and Met-986 (Appendix A, panel 4) during the MD simulation of 100 ns.

Previous in silico studies have also reported hydrophobic interactions with Phe-303, Ile-340, Phe-343 and Gln-990, *π*–*π* stacking with Trp-232, Phe-303 and Tyr-307, and hydrogen bonding with Tyr-310, Gln-725, Glu-875, Tyr-953 and Gln-990 [19,42,45,46], which is in line with the protein–ligand interactions of our lead compounds.

### 3.4. Comparative Interaction Analysis of Lead Compounds with the Selective ABCB1 Dataset

After molecular docking of the entire dataset of 54 compounds (Appendix A) using GOLD 5.3, the binding solutions were analyzed using PLIP [32] and Pymol software [33] as shown in Figure 7.

The frequency and nature of the interaction of different amino acid residues in the binding pocket of ABCB1 are described in Appendix A. This revealed the presence of hydrogen-bond donors (HBDs), hydrogen-bond acceptors (HBAs) and surface contacts (SCs) (hydrophobic and *π*–*π* interactions) between ligands and protein.

PLIF analysis of the larger dataset was therefore broadly consistent with our findings for compounds ‘A’, ‘D’, ‘E’ and ‘F’. The protein–ligand interaction profile of the previously reported classes of selective inhibitors of ABCB1 were compared with the interaction profiles of our identified novel lead compounds and their physicochemical parameters for further lead optimization. The docked solutions of the derivatives of 6,7-dimethoxy-2-phenethyl-1,2,3,4-tetrahydroisoquinoline (Appendix A), galloyl benzamide (Appendix A), and propafenone (Appendix A) exhibiting IC_50_ ranging from 0.33 µM to 10 µM, 0.05 µM to 24.3 µM, and 0.09 µM to 113 µM, respectively, were observed located in the vicinity of transmembrane helices (TM 4–TM 10) as shown in Figure 8. 

It has been previously reported that the introduction of aryl moieties [47,48,49] at the R_1_ position, and amide, ester or alkylamine groups [22] play a dominant role in selectivity and potency of the derivatives of 7-dimethoxy-2-phenethyl-1,2,3,4-tetrahydroisoquinoline. The docking solutions of these compounds (Appendix A) showed that Gln-990 was mainly involved in hydrogen bonding with amide and alkylamine groups (Appendix A), and thus our observation of hydrophobic and hydrogen bonds between the side-chain of Gln-990 and lead compounds ‘A’, ‘D’ and ‘E’ is consistent with the selectivity and activity of these lead compounds in the low nanomolar range. Moreover, most of the interactions, for example in the hydrogen bonding with Tyr-301 and Tyr-307, and hydrophobic interactions with Trp-232, Phe-303, Ile-306, Tyr-307 and Gln-990 with the aryl moiety at the R_1_ position of these compounds (Appendix A) are also evident with our lead compounds (Figure 6).

Protein–ligand interaction analysis of the highly active compound ‘19D’ that belongs to the first subset (derivatives of N-biphenyl-4-yl-3,4,5-trimethoxybenzamide) of the galloyl benzamide derivatives (Appendix A) identified a prominent interaction with Glu-875. Glu-875 was involved in a hydrophobic interaction with the phenyl group of ‘19D’ at the R2 position. This interaction was absent with the least active compound ‘17D’ within the same dataset, due to the occurrence of a bromide group at the R2 position, which is suggestive of its importance for inhibition (Appendix A, panels A and B). The ligand–protein contacts (Appendix A) of lead compounds ‘A’, ‘E’ and ‘F’ preserved that hydrophobic interaction with Glu-875 for the full simulation time, which is again consistent with the high activity of our lead compounds against ABCB1 and the importance of engaging Glu-875.

Overall, the protein–ligand interaction analyses of the lowest and highest activity compounds of the second (3,4,5-trimethoxy-N-(2-nitrophenyl)) and third subset (ethers derivatives) of the galloyl benzamide (Appendix A) revealed the physicochemical properties determining the activities of these compounds (Appendix A, panels C, D, E and F). Pairwise comparisons of the molecular weight, number of heavy atoms, clogP and LipE values (Appendix A) of compounds ‘25D’ and ‘27D’ (from the second dataset) and ‘33D’ and ‘34D’ (from the third dataset) revealed that the difference in their activities might be due to the high clogP and LipE values of ‘25D’ and ‘33D’ as compared to ‘27D’and ‘34D’, respectively.

Protein–ligand interaction analysis of the most and least active compounds of the propafenone derivatives (Appendix A) showed that a high LipE and high molecular weight was consistent with high activity against ABCB1. Our four lead compounds ‘A’, ‘D’, ‘E’ and ‘F’ showed favorable clogP and LipE values in comparison with the dataset of 54 compounds (Appendix A), further strengthening the potential of the lead compounds for the inhibition of the ABCB1. PLIF and SAR analyses of the protein–ligand interactions identified hydrophobic interactions with Leu-65, Phe-303, Ile-340, Glu-875, Met-986 and Gln-990, hydrogen bonds with Ile-306, Tyr-310, Met-986 and Gln-990, and π stacking with Trp-232, all of which can be observed as a stable interaction with our docked lead compounds.

Taken together, the docking and interaction analysis of our lead compounds backed by the ligand–protein contact analysis after MD simulation, and PLIF- and SAR-guided pose analysis of the selective dataset of 54 compounds, has identified crucial protein–ligand interactions. We also observed a direct correlation of clogP, lipophilicity, molecular weight (MW) and the number of heavy atoms with the biological activity of the ABCB1 inhibitors. The physicochemical parameters of the whole dataset (molecular weight, number of heavy atoms, LipE, and clogP) were therefore estimated using Bioloom software (1.5) [50] to identify the most promising drug indices for average oral drug [45,51,52]. The plot between the dock score and pIC_50_ (Appendix A) indicated that the inhibitors (for example, ‘F’ and ‘44I’, ‘A’ and ‘9B’, ‘E’ and ‘54I’) can show a huge difference in their respective activities despite a very small difference in their binding scores, with the major discrepancy observed in the LipE values (Appendix A). Hence, the correlation between the apparent most influential parameter (LipE) and the biological activity of ABCB1 inhibitors was further investigated in a pIC_50_ vs. clogP plot (Appendix A) of the whole dataset along with the first-, second- and third-generation inhibitors (tariquidar, elacridar, and zosuquidar). It was observed that our identified lead compounds ‘A’, ‘D’, ‘E’ and ‘F’ fulfilled the criteria of oral bioavailability with clogP values between 2 and 3 and LipE values greater than 5, as shown in the 3D plot (Figure 9) as proposed by Leeson et al. [51].

To verify the redundancy and collinearity between the pIC_50_, clogP and LipE values of all the compounds of both datasets and to perform the comparative statistical study, principal component analysis (PCA) was also carried out. Both individual compounds and quantitative variables are represented in the PCA diagrams, as shown in Appendix A. The first three principal components (PCs) were associated with eigenvalues >1, while the fourth was equal to 1 and the remaining were <1. Principal component 1 (PC1) explained 63% of variations of the original information, and principal component 2 (PC2) explained 36% of variations, which together exhibited a high level of variation amongst the compounds. The lead compounds ‘A’, ‘D’, ‘E’ and ‘F’ were positively correlated and grouped on the upper right of the plot, however; cyclosporine A showed negative loading on the upper left of the plot due to its lower LipE and clogP values (Appendix A)

### 3.5. Pharmacokinetic Analysis

The drug-likeness and physicochemical parameters of computationally stable compounds ‘A’, ‘D’, ‘E’ and ‘F’ were computed using the online program Swiss ADME. Interestingly, all four leads displayed zero violation against the Lipinski, Ghose, Veber, Egen and Mugge filters, which are used for the evaluation of oral bioavailability for the identification of potential drug candidates. All four compounds also showed high GI absorption with a 0.55 bioavailability score and LD_50_ values of 383 mg/kg, 2580 mg/kg, 2000 mg/kg and 3100 mg/kg, respectively indicative of their non-toxic nature. Moreover, the hits ‘A’ and ‘E’ were classified into toxicity class V, while ‘D’ and ‘F’ belonged to the toxicity class IV; both the classes IV and V are considered good and practically non-toxic.

## 4. Discussion

Overexpression of ABCB1 is a common cause of chemotherapy failure [21,53]. Inhibition of the transporter can readily re-sensitize cell lines to therapeutic drugs in the laboratory; however, translation to the clinical setting so far has failed due to off-target toxicity, lack of selectivity and tissue damage [12]. However, the design of more potent and selective inhibitors remains an attractive strategy to overcome recalcitrant multidrug resistance [12,21,52]. To overcome the obstacles associated with toxicity and off-target effects, we have identified dynamically stable inhibitors through the application of molecular docking and molecular dynamic simulations followed by lipophilic efficiency profiling and pharmacokinetic analyses.

In the present investigation, four experimentally validated lead compounds [21] and a curated dataset of a further 54 compounds from which they were selected were docked into human ABCB1 (6QEX) [15]. The docked lead compounds ‘A’, ‘D’, ‘E’ and ‘F’ interacted with Phe-336 and Met-986, which have been empirically shown to interact with the inhibitors elacridar, tariquidar and zosuquidar via cryo-EM structure determination. Both tariquidar and zosuquidar were also found to interact with Ile-306, Glu-875, Phe-983 and Gln-990, which is consistent with the docking interactions of lead compounds reported herein. Our lead compounds also showed interaction with Leu-65, Trp-232 and Gln-725, which are reported in both the elacridar and tariquidar structures [16]. Some interactions of the lead compounds with Phe-343, Gln-347, Met-986 and Gln-990 have also been observed previously in structures solved with the transport substrates taxol [15] and vincristine [16]. Moreover, molecular docking simulations have previously indicated hydrophobic interaction between inhibitors and the ABCB1 residues Phe-303, Phe-343 and Ile-340, π–π stacking with Trp-232, Tyr-307 and Phe-303, and hydrogen bonding with Tyr-310, Gln-725, Glu-875, Tyr-953 and Gln-990 [15,20,40,52,54,55,56]. Of these amino acids, only Trp-232, Tyr-310 and Tyr-953 are absent from interaction with any of our four lead compounds, although the interaction with Tyr-307 is predicted to be hydrophobic.

Previous studies also suggested that the hydrophobic and aromatic interactions of the inhibitors of ABCB1 are the major contributors of strong binding affinities within the binding pocket of ABCB1 [57]. Moreover, the greater number of these aromatic and hydrophobic interactions contributes majorly to occupying the binding pocket of ABCB1 competitively [58,59,60]. Previously it has been reported that the inhibitors with high clogP and LipE values having at least one tertiary nitrogen atom, aromatic rings, amines, hydrogen bond donors and functional groups like ether, nitrogen, alkyl, carbonyl and arene moieties are crucial for the effective binding of the inhibitors of ABCB1 within the binding pocket of ABCB1 [50,61,62,63].

Some of the previous studies were based on homology models of ABCB1. Here, we have used the cryo-EM structure of human ABCB1 for the docking and MD simulation of lead compounds and the comparison of their binding hypotheses with an extensive dataset of selective and potent ABCB1 inhibitors. The complexes of human ABCB1 with lead compounds ‘A’, ‘D’, ‘E’ and ‘F’ remained in equilibrium and showed stable docking without ligand diffusion for the full length of the simulation. Overall, the apo and the complex state of all four leads and ABCB1 showed overlapping RMSD values after 40 ns till the end of the simulation, which shows the stability of the overall protein complex. Briefly, in the case of lead compound ‘A’, the RMSD of the apo-ABCB1 varied from 2.5 Å to 3.8 Å with an average value of 3.1 Å from 0 to 100 ns. In comparison, the RMSD of the ABCB1-A complex fluctuated within a narrower range of 3.5 Å to 3.8 Å with an average value of 3.6 Å throughout the simulation period of 100 ns. The minor change in the average RMSD of 0.5 from apo- to compound A-bound complexes reflects the overall structural stability of ABCB1 after binding with lead compound A. In the case of lead compound ‘D’, the RMSD of the apo-ABCB1 varied from 2.0 Å to 3.1 Å with an average value of 2.5 Å. In comparison, the RMSD of the ABCB1-D complex varied from 2.4 Å to 3.2 Å with an average of 2.8 Å from 20 ns till the end of the simulation period of 100 ns. This reflects a RMSD difference of 0.4 from the apo to the ABCB1-D complex, which is a very minor difference and does not have a significant impact.

In the case of lead compound ‘E’, the RMSD of the apo-ABCB1 varied from 2.0 Å to 3.4 Å with an average value of 2.7 Å. In comparison, the RMSD of the ABCB1-E complex fluctuated within a narrower range of 3.0 Å to 3.5 Å with an average value of 3.2 Å throughout the simulation period of 100 ns. Thus, a RMSD difference of 0.5 between the apo and the ABCB1-E complex reflects the structural stability of ABCB1 after binding with lead compound E. Likewise, in the case of lead compound ‘F’, the RMSD of the apo-ABCB1 varied from 2.0 Å to 3.3 Å with an average value of 2.6 Å. In comparison, the RMSD of the ABCB1-F complex varied from 3.0 Å to 3.5 Å after 20 ns till end of the simulation with an average value of 3.2 Å. This shows a minor RMSD difference of 0.6 between the apo and the ABCB1-F complex. Similarly, all four lead compounds showed a stable RMSF with negligible fluctuation in the regions of both NBDs. Previous MD simulation studies demonstrated that fluctuations on the order of 1–3 Å can be indicative of stable docking for bimolecular systems [64,65]; however, in the case of larger proteins like ABCB1, a slightly larger fluctuation (4–5 Å) in RMSD is also acceptable [43]. Prior MD simulations of substrates in comparison with inhibitors of ABCB1 showed more fluctuation with the substrates in the RMSF of the nucleotide-binding region as compared to inhibitors [43]. The stable interaction of our lead compounds with minimal induced movement of the NBDs is consistent with inhibition of ABCB1. Coupled with excellent LipE profiles and pharmacokinetic analyses, our four lead compounds ‘A’, ‘D’, ‘E’ and ‘F’ remain promising candidates for the next generation of ABCB1-reversal agents with promising selectivity and potency [49,50].

## 5. Conclusions

Our four lead compounds ‘A’, ‘D’, ‘E’ and ‘F’ were not only found to inhibit ABCB1 with high potency to overcome MDR in cell culture [21], but also displayed interaction patterns within the binding pocket of human ABCB1 that are similar to the selective and potent dataset of the inhibitors of ABCB1. These lead compounds showed stable interactions with minimum fluctuations in RMSD and RMSF during MD simulations and also showed ideal pharmacokinetic properties, LipE and clogP values, which will help to realize their potential as candidate inhibitors of ABCB1.

## Figures and Tables

**Figure 1 biomolecules-14-00114-f001:**
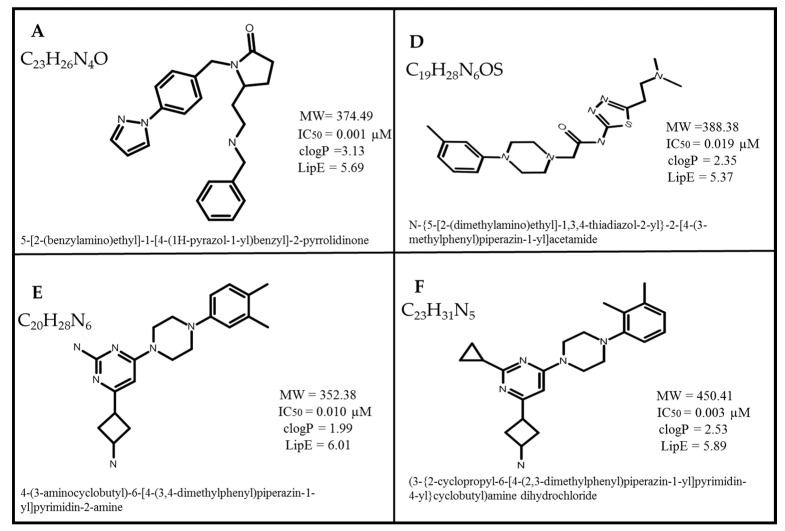
2D structures of the lead compounds ‘A’, ‘D’, ‘E’ and ‘F’ along with their chemical formula, name, molecular weight, clogP, IC_50_ for inhibition of calcein-AM transport and LipE values [21].

**Figure 2 biomolecules-14-00114-f002:**
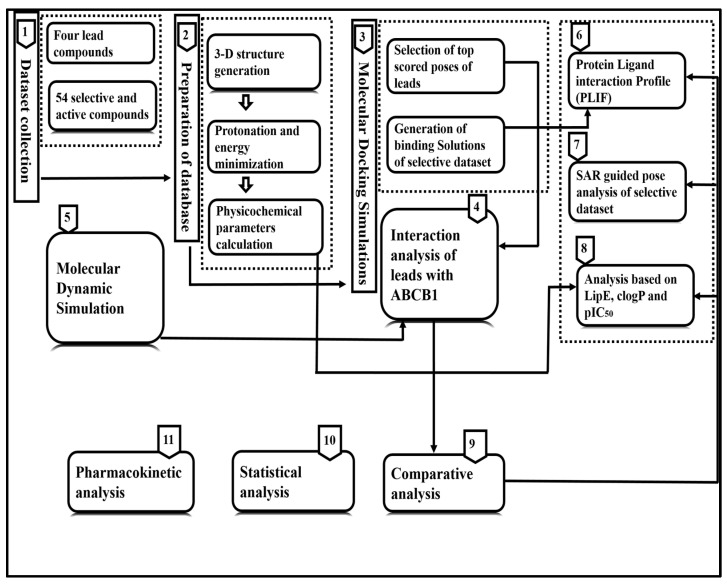
Detailed workflow of the computational methodology adopted to probe the 3D features of ligands of ABCB1. The dataset of 58 inhibitors was docked into the binding pocket of human ABCB1 (6QEX), followed by ligand–protein interaction analysis, SAR-guided pose analysis, PLIF, IF, statistical analyses and molecular dynamics simulation for 100 ns. Finally, pharmacokinetic analyses of the selected inhibitors was performed to identify likely non-toxic and efficacious inhibitors of ABCB1.

**Figure 3 biomolecules-14-00114-f003:**
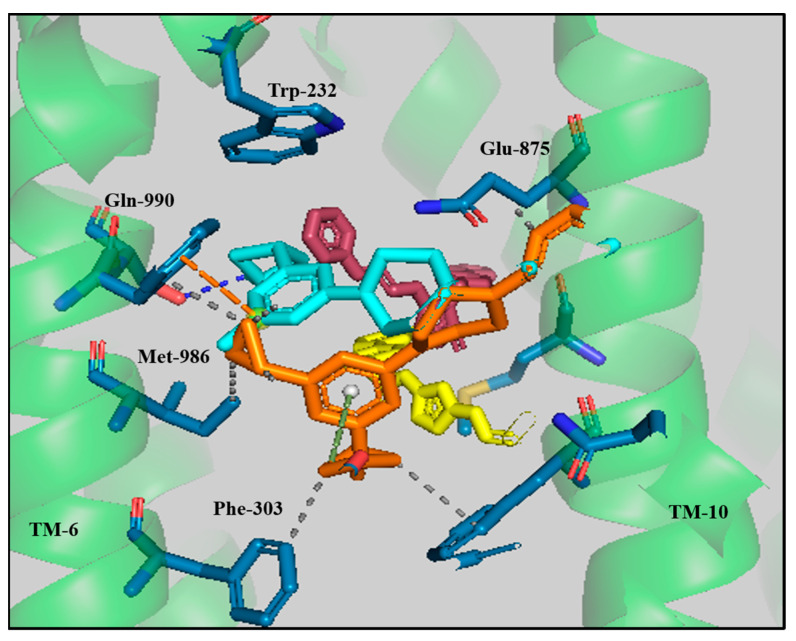
Binding solutions of the lead compounds ‘A’ (maroon), ‘D’ (yellow), ‘E’ (cyan) and ‘F’ (orange) within the binding pocket of ABCB1 (6QEX). The interacting residues are shown in stick form. Green dotted lines represent π-stacking, grey dotted lines show hydrophobic interactions and blue dotted lines represent hydrogen bonding.

**Figure 4 biomolecules-14-00114-f004:**
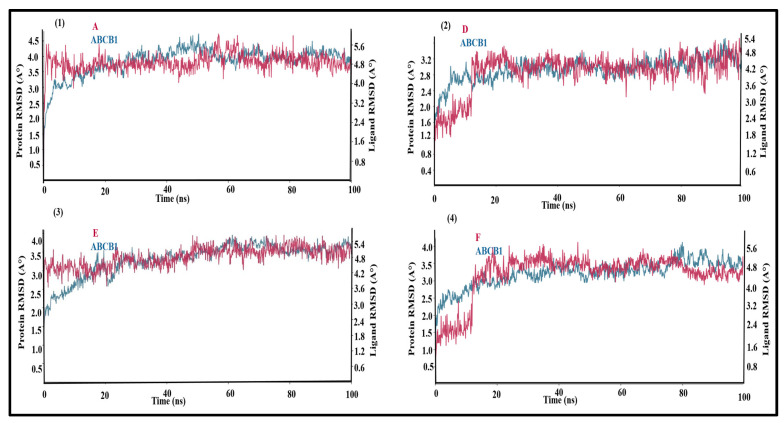
RMSD vs. time (100 ns) of the simulated ABCB1–lead complexes with protein-RMSD displayed in blue while the RMSD of the ABCB1–lead compounds is displayed in maroon. (**1**) RMSD of ABCB1 and ‘ABCB1-A’; (**2**) RMSD of ABCB1 and ‘ABCB1-D’; (**3**) RMSD of ABCB1 and ‘ABCB1-E’; (**4**) RMSD of ABCB1 and **‘**ABCB1-F’ complexes.

**Figure 5 biomolecules-14-00114-f005:**
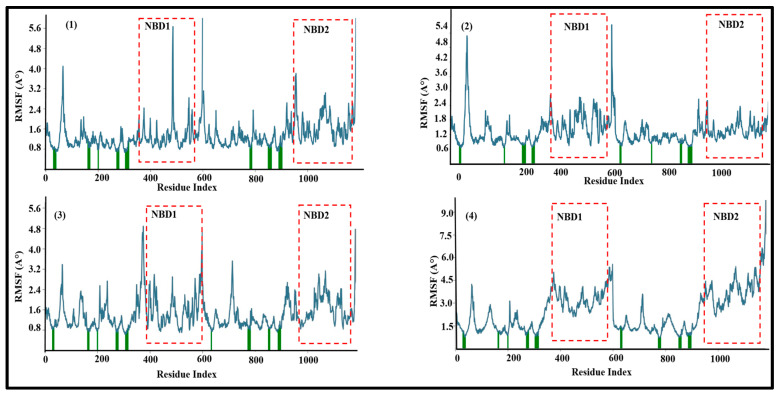
RMSF for the simulated ABCB1–ligand complexes for 100 ns. (**1**) RMSF of ‘ABCB1-A’; (**2**) RMSF of ‘ABCB1-D’; (**3**) RMSF of ‘ABCB1-E’; (**4**) RMSF of ‘ABCB1-F’. Residues that interact with the compounds are marked with green bars.

**Figure 6 biomolecules-14-00114-f006:**
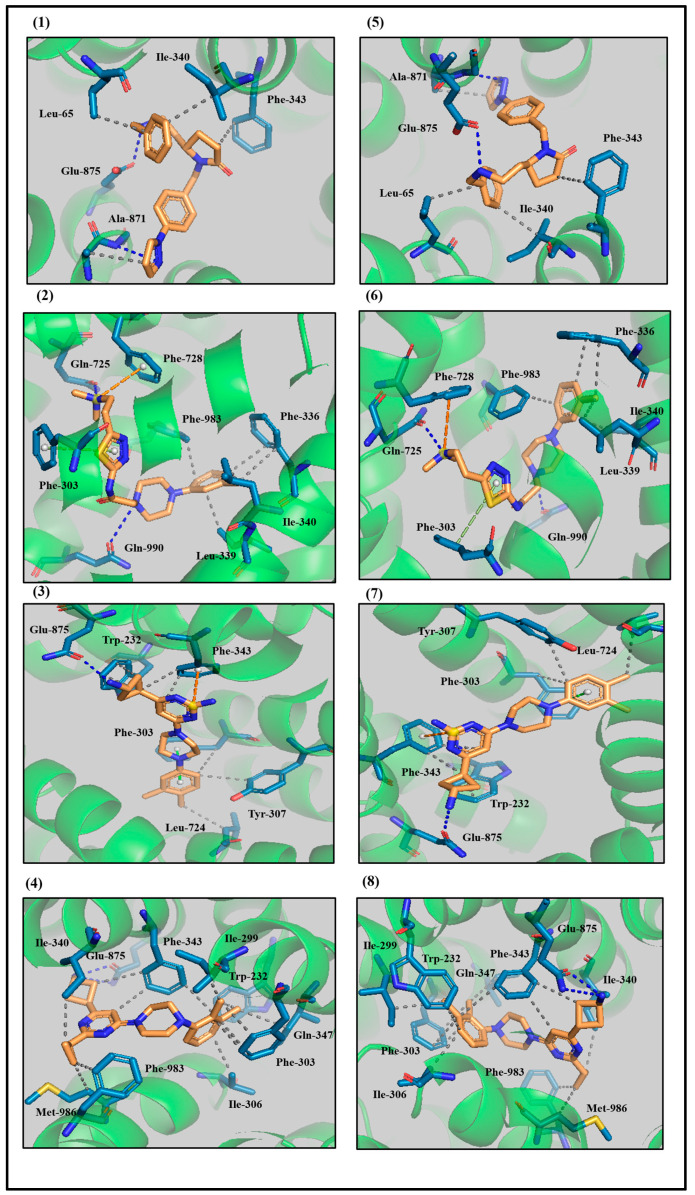
Comparison of ligand–protein interactions of lead compounds complexed with ABCB1. (**1**–**4**) Interaction pattern of leads ‘A’, ‘D’, ‘E’ and ‘F’, respectively in the binding pocket of ABCB1 immersed in a full membrane-aqueous environment at 0 ns. The interacting residues along with the lead compounds are shown in the stick form and hydrophobic, hydrogen bonding and *π*-stacking interactions are depicted by grey, blue and green dotted lines, respectively. (**5**–**8**) Interaction pattern of lead ‘A’, ‘D’, ‘E’ and ‘F’, respectively in the ABCB1-binding pocket, which was entrenched in a complete membrane-aqueous system after 100 ns.

**Figure 7 biomolecules-14-00114-f007:**
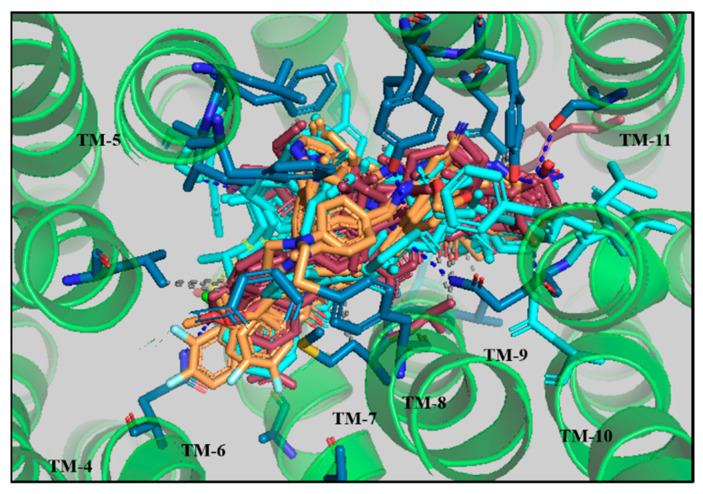
Binding solutions of the derivatives of 6,7-dimethoxy-2-phenethyl-1,2,3,4-tetrahydroisoquinoline (orange), galloyl (raspberry), and propafenone (cyan) within the binding pocket of human ABCB1 (6QEX). The interacting amino acids are displayed in stick form (blue).

**Figure 8 biomolecules-14-00114-f008:**
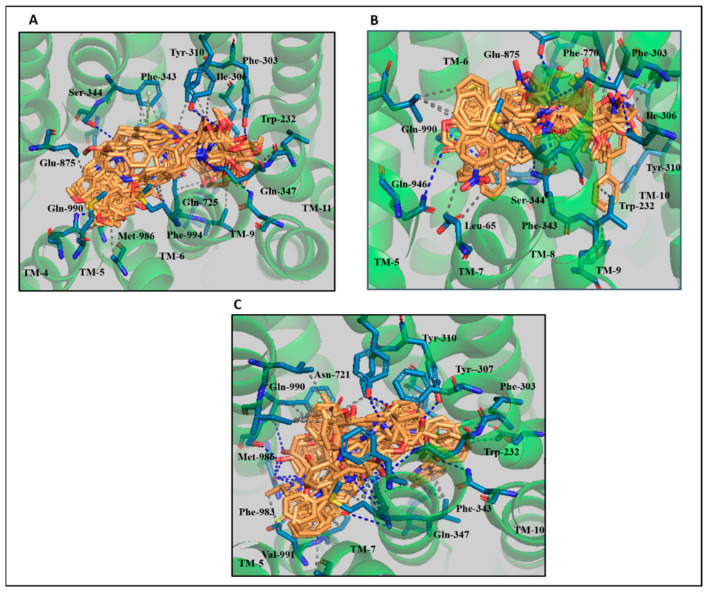
(**A**) Protein–ligand interaction pattern of binding solutions of the derivatives of 6,7-dimethoxy-2-phenethyl-1,2,3,4-tetrahydroisoquinoline and 6QEX. (**B**) Binding solutions of the derivatives of the galloyl showing interactions in the binding pocket of ABCB1. (**C**) Binding solutions of the propafenone derivatives showing interactions with ABCB1.

**Figure 9 biomolecules-14-00114-f009:**
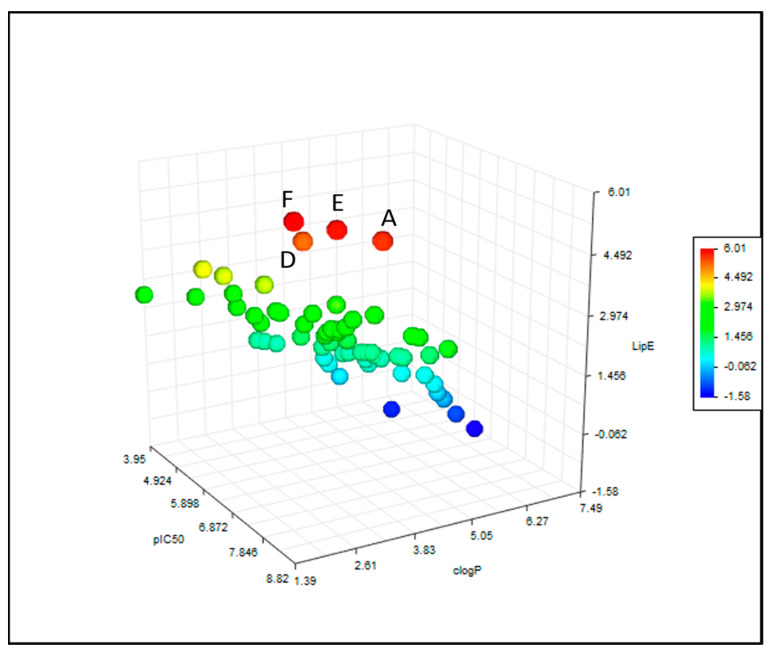
Selective inhibitors of ABCB1 and potential leads are represented as points in 3D space mapped according to activity values, molecular weight and clogP values (points are color-coded according to the PIC_50_ values). The ellipse shows our lead compounds (red color of leads is for the representation of LipE > 5).

## Data Availability

Computational datasets, docking and MD simulation files are available upon request from Dr. Ishrat Jabeen.

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
