# Peer review of "Molecular Modeling Studies to Probe the Binding Hypothesis of Novel Lead Compounds against Multidrug Resistance Protein ABCB1"

_biomolecules, 2024, doi:10.3390/biom14010114_

Round 1

Reviewer 1 Report

Comments and Suggestions for Authors

From Reviewers

Comments and suggestions for authors

The manuscript by Cheema et al. extends upon their previous study (Cheema Y., et al., 2023) to further scrutinize the properties of four lead compounds exhibiting inhibitory effects of ABCB1 which is a significant player among the three major ABC transporters responsible for multidrug resistance (MDR) in human cancers, sharing a broad spectrum of substrate/inhibitor specificity with the other two transporters, ABCG2 and ABCC1. The study primarily employs computational approaches, such as docking, MD simulations, and collective datasets from previous literatures for calculations.

Concerning the intricate mechanism of ABC transporters that requires ATP hydrolysis to drive conformational change and regulate specific substance efflux. Therefore, relying solely only on computational analyses may not sufficient to draw conclusions or interpretations regarding whether the candidate compounds are suitable and provide the same activity as predicted from computational analysis under real physiological conditions. Notably, the study needs the appropriate approach with suitable biochemical or in vivo experimental validations to ensure a comprehensive understanding and facilitate an informative discussion.

The manuscript is criticized for not contributing novel knowledge, relying on previously published data, and lacking validation experiments to support their computational predictions. The major concerns in this manuscript are highlighted including non-novelty, study flow, sufficient results, comprehensive discussion, reasonable calculations, and overall figure quality. The analysis is criticized for lacking coherence, presenting some inconsistent results, and providing unclear conclusions. Additional validation experiments should be included by the author to investigate the potential of these four lead compounds as potent novel drugs for ABCB1. The discussion should be intensively re-written and improved. Many typos and mistakes are detected in this manuscript. Therefore, the author should pay more attention to carefully check through the whole manuscript again.

There are many important or missing points that have to be improved. Please follow the major and minor concerns below:

1.     The title does not accurately represent the scope of this work and should be modified because the data did not provide the 3D structural profile of lead compounds or ABCB1 that are crucial for their potential efficient. The author does not discuss or provide any information about the structural profile of the compounds that are essential for the drug potential. Instead, solely represents computational prediction and calculation based on existing data, lacking any appropriate biochemistry analysis or validation. A more fitting might be “Computational analysis of potential lead compounds for multidrug resistance protein ABCB1”

2.     I totally agree with the statement in lines 40-42, emphasizing that antibody-bound structures may not accurately reflect the proper conformation, including both shape and side chain orientation around the binding pocket. Due to the interference or potential inhibition of transport function and ATPase activity by fab-binding. However, in this study the authors used 6QEX which is an ABCB1 in complex with UIC2 fab and taxol, for all docking and MD simulations analysis. The rationale behind choosing this structure, despite the potential inappropriate effects from fab-binding, is not clearly explained. It would be prudent for the authors to justify this choice and more importantly to explore alternative fab-free structures. Conducting docking and MD simulations with such structures and subsequently comparing the results could provide valuable insights into the drug interaction at the binding pocket.

3.     Figure 3 depicting the overlapping of compounds A/D/E/F at the same binding pocket is too dense and is difficult to distinguish each individual compound in the figure. I would suggest presenting them as separate individual figure for each compound, highlighted with distinct color for clarity. Additionally, the figure legend should be written properly with more detail regarding the color code and representative styles to enhance comprehension.

4.     The MD simulations of four lead compounds with ABCB1 in Figure 4 reveal relatively high RMSD values for both compounds and protein. Could the author provide the numeric data of the RMSD values of each compound individually in the text? Notably, the results suggest a significant evaluation in RMSD upon the binding of each lead compound, from the apo state to the complex-bound state: compound A (3.5 Å in apo state to 5 Å in complex of ABCB1 with A), D (3 Å to 4.5 Å), E (3.5 Å to 5 Å) and F (3.5 Å to 4.8 Å). Do these results indicate that binding of these four lead compound reduce stability? Typically, the binding of an inhibitor should enhance protein rigidity and stabilize the protein complex. Therefore, please elucidate this finding and write in the discussion.

In addition, RMSD value indicates the stability of protein-complex which depends on the binding interaction and energy between protein-ligand. The optimal RMSD value should be less than 2 Å, while 3 Å is also acceptable. However, if the RMSD is > 4 Å, this is considered unacceptable or might signify an issue! Why does the author consider that RMSD > 4 Å is still good for interpretation, especially when the references that the author refers still recommend at below 3 Å.

5.     In Figure 5, the RMSF of ABCB1-ligand indicates that binding of these four lead compounds does not exhibit fluctuations in the NBDs. Does this result indicate that all four compounds inhibit NBD dimerization and block ATP hydrolysis? In order to verify this finding, ATPase assay is needed to validate this simulation which will not only help to confirm the analysis but also providing insights into the molecular action of the compounds on the activity of ABCB1. Please include this experiment and discuss this point in the discussion section.

6.     The results in section 3.3 do not really provide consistent data and are quite confusing. Importantly, there are several inconsistences between the results in Figure 6, 7 and Table 1.

·         Regarding Figure 7, there is a need for clarification on how to calculate and interpret the interaction fraction %? Please explain and write down in the Material and method section. This is particularly crucial as the interacting residues in the figures do not correspond to those in Table 1 and text.

·         For example, in lines 267-268, “all the four lead compounds showed high levels of interaction with Phe-343 …” But in Table 1 indicates that compound D does not interact with Phe-343. The discrepancy raises questions about why Figure 7 suggests that compound D has a high interaction with Phe-343.

·         Similarly, in lines 269-270, “compounds A, E and F showed high percentage of interaction with Ile-306 and compound D, E and F showed interaction with Glu-875”. However, Table 1 contradicts this information, indicating that Ile-306 interacts only with F, while Glu-875 interacts only with E.

·         The author should provide an explanation for these inconsistencies to ensure clarity and coherence in the interpretation.

7.     There are many mistake in Result section 3.3

·         In line 266, Tyr-310 is not in any table 1 nor Figures 6-7. Is it wrong and should be Tyr-307?

·         There are two typos in the Figure 7 at the x-axis label. First, “Interactiong” should be “Interaction” or “Interacting”.

·         And the second typo, “amini” should be “amino”.

All in all, I would recommend removing section Results 3.3, Table 1, Figure 6 and 7. Or move them to supplementary part (after clarify all points).

8.     The color codes in Figure 8 are confusing, as the Olive color is not discernible, and the figure legend does not mention the orange and cyan colors. It would be beneficial if the author could modify the figure legend to accurately represent the colors used in the figure. Additionally, it is advisable to providing more discussion about pattern of binding result of Figure 8 in the discussion section.

9.     Figure 9 is also very confusing. How did the author calculate to derive these results and how to interpret? Please provide detailed information in “Materials and methods” section and provide exact data as table for the results. The figure legend is also incomplete; what do the colors of green and blue mean? Please provide a clear figure legend and elucidation on the method for analyzing this data.

10.  There is incorrect information in line 304 because propafenone should be in supplementary table 4 not 3. Please check it!

11.  Figure 10 show that compounds within the same structural category share the same binding pocket. Do they also share the same interacting residues? However, each individual compound has a significantly different with a broad range of inhibitory efficiency (IC50). Therefore, what are the crucial factors essentially indicating the inhibitory effect? How can one discern and extract crucial interactions, side chains, functional groups, etc. The author should delve into the detail of each category’s interaction that may influence to the activity and please provide a detailed discussion on this matter.

12.  Figure 11 lacks novel informativeness as the relationship among LipE, pIC50, and cLogP can be easily represented through a straightforward simple equation, such as “LipE = pIC50 – cLogP”, without any additional complicated factors. Therefore, the fancy 3D representation of this simple equation seems unnecessary. Additionally, using PCA to identify collinearity between LipE, pIC50, and cLogP, as clearly demonstrated in the simple equation, may be excessive for a straightforward relationship involving these three known factors. It appears that this figure does not contribute novel information but rather presents similar results to those already published in previous studies, albeit in a different visual format. Figure 11 should be labelled properly, at least all four compounds and some example per group color.

13.  Lines 392-399, should be moved to the end of Result section 3.2, and Figure 12 should be moved to supplementary information.

14.  In Figure 12, why MD simulations of 35E showed the shifting of ligand RMSD at ~55 ns? Please explain in discussion.

15.  Where are the texts explaining the results in Figure 12 B, C, E and F?

16.  Line 395, “35I” is incorrect. It should be “35E”

17.  Lines 426-442 should be rewritten with more conclusive information, incorporating analysis and discussion to summarize the overall analysis and at least attempt to elucidate the important interaction essential at the binding pocket. For instance, the author should address questions such as: what are the common interacting residues for high potent inhibitors? What types of interactions are considered crucial and what is the quality of interactions required, among other relevant points? Etc.

18.  Line 450-452 state that an RMSD of 1-3 Å is acceptable, but all the results are > 3 Å. What does this mean? Can we accept these >3 Å?

19.  Line 463, it is wrong reference. (Not 18 but should be 21)

20.  Lines 465-468 need modification because relying only on computational analysis predictions by the high value RMSD and with only RMSF are insufficient to conclude the stable interactions of these four compounds in physiological environment. Since the actual biological conditions are more complex, with numerous additional factors influencing drug efficiency. To draw confident conclusions, additional extensive biochemical validation experiments are crucial. The additional experiments for instance, inhibitory tests of transport functions with different substrates (considering ABCB1’s broad substrate range overlapping with ABCG2 and ABCC1). Mutagenesis is necessary to characterize specific interacting residues and function. ATPase assays would reveal the specific inhibition mechanism. Assessing toxicity in cell-based studies, or ideally in vivo using proper organism models, is very useful. Notably, this manuscript lacks any experimental validation to confirm its computational predictions.

21.  All data in Supplementary information are not generated in this study but collected from previous papers. Therefore, it need to be cited properly in every table as well as indicated in “Materials and Methods”.

22.  Then nemoclature of the compound name in the supplementary of this manuscript is totally difficult to match with the original paper. Therefore, they should be improved, for example, indicate in the legend or add reference accordingly.

23.  I am confident that the IC50 values of CsA, Tariquidar, and Elacrida for ABCG2 and ABCC1 are already available in published literatures. It would be better to assess the specific inhibitory effect on ABCG2 and ABCC1. The author should include this missing information in the supplementary tables 1.

The followings are the minor points:

24.  The chemical structure of compounds D and F in Figure1 are not represented in a proper ration. Could the author correct them?

25.  The equations for RMSD (line 146) and RMSF (line 153) were not correctly written, that altering the calculating meaning of results. If the author is unable to type the proper equations, I would recommend adding brackets around 1/N and 1/T in its equation.

26.  Font size of all labelling graphs in Figures 4, 5, 11 and 12 should be increased for clarity.

Comments on the Quality of English Language

The grammar in this manuscript requires careful checking for typos again.

Author Response

We are very grateful to the reviewer for the constructive suggestions. A point by point response to the reviewer's comments is attached.

Reviewer 2 Report

Comments and Suggestions for Authors

The paper titled “Probing the 3D Structural Profiles of Potential Leads of Multi-drug Resistance Protein ABCB1” by Yasmeen, et al. investigates the conformational dynamics of the drug efflux pump ABCB1/P-glycoprotein (P-gp) induced by the binding of potential inhibitors using molecular dynamics simulations. The manuscript presents compelling evidence for the potent and selective inhibition of ABCB1 by four lead compounds, revealing insights into their binding interactions and stability within the protein structure.

The manuscript holds potential value, but its presentation requires improvement. To consider its publication, I request the authors to address the following points.

Major Concerns:

1.     Structural Comparisons: While the 2D structures suggest similarity between compounds E and F versus A and D, the RMSD results indicate similarities between D and F, and the RMSF shows similarities between D and E. Please explain these discrepancies. And elaborate on the role of the cyclopropyl group in compound F.

2.     Methodology Clarification: Specify membrane components in the molecular dynamic simulation method, including the amount of POPC, simulation box size, and total atom numbers used.

3.     Region Definition: Include a schematic depicting the ABCB1 domain organization and sequence locations referred to in Line 225 to aid readers in locating mentioned regions.

4.     Simulation Duration: The stability of the compounds over 100ns appears promising. However, a longer simulation, such as 500ns, would provide valuable insights into the stability over time. Consider extending the simulation duration for a comprehensive stability assessment.

5.     Data Presentation: Table 1, containing ligand-protein interaction distances, could be moved to supplementary materials. Instead, consider a figure illustrating changes in side chain interactions, offering additional insights beyond what's represented in Figure 6.

6.     Clarify whether all interactions are included in Table 1 and why there were no changes observed.

7.     Definition of "Interaction Fraction": Provide a clear definition of "Interaction Fraction" to enhance reader understanding and facilitate interpretation of the results.

8.     The workflow depicted in Figure 2 is challenging to navigate, making it difficult to establish connections between the smaller figures and the corresponding text. Simplification and clarity in the figure layout are recommended for better comprehension.

9.     Figure Presentation: The simultaneous overlay of the four lead compounds in Figures 3, 8, and 10 complicates differentiation and analysis. I recommend presenting separate figures for each compound, allowing for clearer comparison and analysis of individual interactions with ABCB1.

10.  Figure 5 Insights: Explain the observed significant peaks in (1)NBD1 and (4)NBD2 compared to others in Figure 5, offering a detailed rationale for these differences.

11.  Missing Video: Please ensure that the mentioned video is included.

Minor Concerns:

1.     Citations: Provide citations for "Desmond" in Line 264 and for the statement regarding chemotherapy failure in Line 417.

2.     In Line 462, the term "Strikingly" should be omitted as the coordination observed is not deemed striking; rather, it's derived from the cell results (Int. J. Mol. Sci. 2023) and simulations presented in this paper.

Author Response

We are very grateful to the reviewer for the constructive comments and suggestions, a point by point response to the reviewer's comments is attached.

Reviewer 3 Report

Comments and Suggestions for Authors

Synopsis:

In this study, Cheema et al aimed to identify the structural basis of inhibitor binding to P-glycoprotein (Pgp/ABCB1). The authors performed extensive molecular docking and molecular dynamics simulations. Related to the authors’ previous works, the authors looked into 54 compounds from previous studies (ref 23 & 24), categorized them based on these A/D/E/F classes, and evaluated their predicted indices in ligand-protein interaction, structure/activity relationship, ligand interaction fingerprints, and interaction fraction.

Strength:

This is a follow-up of the authors’ previous studies (PMID: 36982374). This manuscript summarized a list of residues that interact with the four curated substrate types, namely A, D, E & F. This finding is novel and the main strength of this study.

Major concerns:

1. This manuscript describes results exclusively from computer-aided prediction while citing the functional data from this previous study. There is no functional assay in this study to validate the impact of compound-interacting residues on Pgp’s functionality. For instance, a drug transporter assay for tested compounds.

2. While nice to see a list of interacting residues, at least an in vitro functional validation by site-directed mutagenesis, for example, on these residues would be necessary in this follow-up study. Or can the authors find existing data to support such a prediction?

3. The IC50 for the selected 54 compound leads is in the 𝜇M to sub-𝜇M range. This is much higher than A/D/E/F, as used by authors in the previous studies (PMID: 36982374). Are these tested by an in vitro functional assay, such as ATPase using purified proteins or drug efflux using cell models?

Minor suggestions:

1. Line 50: lipophilic efficiency (LipE) was already defined earlier. Use of the abbreviation is ok here.

2. Figure 3 is very busy. For example, it is not clear what the 4 compounds are and they are jammed to each other.

3. Bigger labels and font sizes for Figures 4 & 5.

4. Line 339: “… despite”. Despite what?

5. Line 365: Interaction Fraction (IF) was already defined earlier.

6. Line 443 states “Most of the previous studies were based on homology models of ABCB1.” This is an overstatement. Before cryo-EM data, there have been several rodent Pgp crystal structures, which share high sequence identity to human proteins. The wealth of biochemical studies over the past 3 decades supports the use of non-human proteins, such as mouse Mdr3.

Comments on the Quality of English Language

The English is read well in this manuscript.

Author Response

We are very very grateful for the constructive comments of the reviewer. A point by point response of the reviewer's comments is attached. 

Round 2

Reviewer 3 Report

Comments and Suggestions for Authors

The authors addressed and clarified concerns from the previous review, and the manuscript has improved significantly. No further comments from this reviewer.